# Decoding past microbial life and antibiotic resistance in İnonü Cave's archaeological soil

Sukran Ozturk[1]☯*, F.Gülden Ekmen[2]☯, Hamza Ekmen[2], Esra Mine Ünal[3,4], Ayşegül Er[3,4], Emre Keskin[3,4,5], Benjamin Stanley Arbuckle[6]

1 Zonguldak Bülent Ecevit University Faculty of Pharmacy Pharmaceutical Microbiology, Zonguldak, Türkiye, 2 Zonguldak Bulent Ecevit University Teoman Duralı Faculty of Humanities and Social Sciences, Zonguldak, Türkiye, 3 Ankara University, Agricultural Faculty,Department of Fisheries and Aquaculture, Evolutionary Genetics Laboratory (AGL), Ankara, Türkiye, 4 Agrigenomics Hub (AgriGx) Animal and Plant Genomics Research Innovation Centre, Ankara, Türkiye, 5 Ankara University Aquaculture Research and Application Center (ASAUM), Ankara, Türkiye, 6 University of North Carolina at Chapel Hill, Department of Anthropology, Chapel Hill, North Carolina, United States of America

☯ These authors contributed equally to this work.
* sukranozturk79@gmail.com (SO); ekmengulden@gmail.com (FGE)

## Abstract

This study, which bridges the disciplines of archaeology and microbiology, examines the ancient bacterial communities and antibiotic-resistance genes in soil samples collected from İnönü Cave in Zonguldak, Turkiye. Our aim is to provide a comprehensive understanding of historical human activities and their influence on microbial communities. Soil samples were gathered from four distinct cultural levels from the Chalcolithic Age to the Early Iron Age. The microbial communities were characterized, and antibiotic-resistance genes were identified using high-throughput sequencing of 16S rRNA genes and metagenomic studies. This interdisciplinary approach not only enriches our understanding of ancient microbial communities but also opens up new avenues for research and collaboration. The results of our study showed a wide range of microorganisms, including prominent bacterial groups such as Acidobacteriota, Actinobacteriota, Bacteroidota, Chloroflexi, Cyanobacteria, Firmicutes, Myxococcota, and Proteobacteria. The study identified the presence of the tetracycline resistance gene tetA in Chalcolithic samples, the class 1 integron intl1 in Early Bronze Age samples, and the oxacillinase gene OXA58 in Late Bronze Age samples. These findings underscore the long-term impact of human activities on microbial communities, as antibiotic-resistance genes have been present and have remained over various historical periods, perhaps influenced by both human activities and environmental variables. This knowledge is crucial for understanding the resilience and adaptability of microbial communities in the face of human-induced changes. The coexistence of these resistance genes and alterations in the microbial population suggest substantial connections between human activities and soil microbiota. This study, which draws on the fields of archaeology, microbiology, and environmental science, offers valuable

**Data availability statement:** The datasets generated and/or analyzed during the current study are available in the NCBI BioProject repository, BioProcecj ID: PRJNA1134133.

**Funding:** ZBEUN Scientific Research Coordinator (BAP) number 2021-91149634-02. The funders had no role in study design, data collection and analysis, decision to publish, or preparation of the manuscript.

**Competing interests:** No authors have competing interests Enter: The authors have declared that no competing interests exist.

insights into the ancient microbial ecology and underscores the enduring presence of antibiotic resistance. It emphasizes the necessity of a comprehensive, interdisciplinary approach, spanning multiple fields, to comprehend microbial communities' evolution and resistance mechanisms in archaeological settings.

## Introduction

The ruins and finds unearthed in archaeological contexts reflect the results of all the vital dynamics and events related to past human production and consumption processes. These dynamics and events include the development of agricultural practices, the emergence of trade networks, and the evolution of dietary habits. Archaeological artifacts (metals, ceramics, glass, etc.) and organic remains (animal bones, seeds, microorganisms, etc.) represent complex assemblages within their depositional context, and this diverse body of information can reveal data that concerns many disciplines [1].

The field of ancient DNA (aDNA) and microbial communities in archaeological contexts has received considerable attention in recent years. These investigations offer a distinct viewpoint on human behavior, health, and interactions with the environment in the past. By examining microbial DNA found in archeological sites, scientists can extract significant insights regarding the microbiomes of ancient civilizations, their food patterns, and the occurrence of contagious illnesses [2]. Furthermore, examining ancient microbial communities allows us to comprehend the historical influence of human actions on environmental microbial ecosystems. Recent interdisciplinary studies have shown that the materials contained in archaeological sediments provide information beyond just defining culture [3]. In this context, organic remains dating back thousands of years play an important role. While research on organic remains found in archaeological sites has gained momentum in recent years, the number of studies, especially on microbial diversity and change, still needs to be increased. High-throughput sequencing technologies have opened new frontiers in microbial community analysis and have become widely used in assessing the diversity of bacterial components in soil [4]. The recent application of next-generation sequencing methods such as Illumina and Roche provides reliable and accurate results in detecting microbial taxa. 16S rRNA studies are preferred as an excellent phylogenetic method to obtain important information about investigating and detecting microbial taxa and antibiotic resistance genes in soil samples from archaeological excavations [5,6].

Now, more than three decades later, scientists and researchers are actively engaged in ongoing research to solve various puzzles related to the origin of humanity, migration patterns, the emergence of infectious diseases, and their spread among ancient populations. aDNA analysis has emerged as a state-of-the-art genetic tool gaining momentum worldwide, changing our comprehensive understanding of the past. For example, in a recent groundbreaking study, aDNA analysis of seven individuals revealed the origin of the plague strain that caused the Black Death in present-day [7,8].

Today, with the help of aDNA, the current atlas of genetic diversity is no longer limited to a glimpse of the diversity observed in modern populations across the globe. Instead, it is continuously updated with datasets that track changes in the genetic origins of human, animal, plant, and even microbial populations as they grow, implode, and adapt to new environmental factors [9,10]. Beyond its biological and archaeological prospects, aDNA also holds the potential to establish strong and intriguing political and cultural links with other nations [11]. More broadly, aDNA data have revolutionized our knowledge and curiosity and have led to the publication of an enormous amount of literature. aDNA from microorganisms, including pathogens and commensals, can provide insight into human health and changes in the diet and ecology of diseases. Initially, research on ancient microbes used PCR-based techniques to identify pathogens. However, PCR-based techniques were less successful than Next-Generation Sequencing (NGS)-based techniques in distinguishing between ancient and modern contaminant microbial DNA [12]. NGS-based techniques refer to a set of advanced DNA sequencing methods that allow for the rapid and high-throughput sequencing of DNA, providing a more comprehensive understanding of the genetic makeup of ancient microbes.

The introduction of high-throughput sequencing tools has completely transformed the discipline of microbial archaeology. Next-generation sequencing (NGS) methods offer a complex understanding of the diversity and structure of bacterial taxa, enabling the comprehensive study of microbial communities. This method is an important technique in fields such as agriculture, ecology and human health to determine the microbial community structures of environmental samples. Deep sequencing and the capacity to sequence multiple samples make metagenomic sequencing technologies highly attractive for investigating microbial species diversity [13–15]. Microbial communities in soil participate in various ecological interactions between organisms and in biogeochemical processes of nutrient mobilisation, decomposition and gas fluxes (Urbanova, Therefore, metagenomic studies of soil communities are crucial for understanding these processes [16]. Determination of microbiota in prehistoric soil samples by metagenomic analyses to reveal and interpret these processes provides valuable data both to shed light on that period and to make a comparative evaluation with today.

Methods such as 16S rRNA gene sequencing have become indispensable for detecting and characterising microbial populations in many contexts, including ancient sites. Technical advances have greatly improved our capacity to identify and study ancient microbial DNA, allowing scientists to investigate complex microbial interactions and their evolutionary pathways spanning millennial timescales [17]. Studies on soil samples obtained from defined fills of archaeological sites have a long history. Most of these studies focus on general aspects of soil chemistry, particularly pH variation, microelements, or micromorphology. However, soil studies within the scope of archaeological projects in recent years have brought a microbiological perspective to investigating paleosols and sediments affected by anthropogenic factors [18,19].

Microbiology provides valuable data that complement and enhance knowledge of past events as a tool in non-biological fields. For example, microbiological data at the geological scale can contribute to understanding and reconstructing past climatological, environmental, and sedimentary data. Moreover, microbiological data can contribute to discovering human cultural habits throughout history, diseases associated with microorganisms, and changes in cultural artifacts due to biochemical reactions [20].

Human activity affects soil properties and is accompanied by the introduction of specific organic matter into the soil. After removing the anthropogenic influence, soil microorganisms lose organic matter due to mineralization and transformation. However, the former anthropogenic influence on soils can be preserved in the soil microbiota and their activities [18,21]. High organic matter input usually stimulates microbial activity, which leads to increased microbial biomass and enzyme activity. There is evidence that paleosols around archaeological monuments, such as the settlement layers of ancient settlements, carry a record of the past's environmental conditions and the life of ancient people [19,22]. This study, which examines the local soil bacterial communities using metagenomic analysis of soil samples from the İnönü Cave in the Zonguldak province of Türkiye, makes a significant contribution to the understanding of ancient microbial communities and antibiotic resistance. The study spans from the Chalcolithic Age to the Early Iron Age and focuses on testing for the presence of antibiotic resistance genes using molecular methods. It also investigates potential diseases and treatment

methods. The research aims to examine the microbial communities and antibiotic resistance genes in soil samples from an archaeological site, revealing the influence of past human activities on the diversity of microorganisms and the presence of genes that confer resistance to antibiotics. By integrating archaeology and microbiology, this multidisciplinary approach offers a unique opportunity to understand the historical and environmental factors that shaped ancient microbial populations. The study adds to the growing body of knowledge on ancient microbiomes and provides valuable insights into the persistent presence and evolution of antibiotic resistance in (pre-) historic settings.

## Site (İnönü Cave) and samples

İnönü Cave is a cave of volcanic origin located in the Cambu geological formation. This volcanic cave, located on the Western Black Sea coast of Turkiye, developed in several stages (Fig 1) [9]. Its formation began with the northward subduction of the northern Neo-Tethys ocean, which produced magma due to partial melting and migration of this magmatic body to the surface, forming a typical volcanic arc, part of the Campanian upper magmatic succession (Kökyol, Unaz, Cambu Formations) (Fig 1) [23].

İnönü Cave, located in the valley formed by the Gülüç Stream flowing into the Black Sea, overlooks an area suitable for agricultural and livestock activities. The cave has three interconnected chambers: The cave's width, with its mouth facing west, reaches approximately 25 m in the interior, and its height reaches 10 m. Inside the cave, which is well-lit from sunrise to sunset, there is a natural spring water source still drinkable today. Due to all these features, İnönü Cave was chosen as a living space by people in prehistoric and early historical periods (Fig 2) [24].

Animal remains identified in the stratigraphic sequence at İnönü Cave reveal a combination of an animal economy strongly adapted to the Pontic region's rugged terrain and forested environment and long-term continuity in cultural preferences (Fig 3) [26,27]. Mammals dominate the animal community, although small numbers of tortoise, fish, bird, and crab remains have also been identified (Table 1). Among mammal remains, interestingly, suids are the most abundant (36% of specimens identified by genus), followed by caprines (goats and sheep 21%), deer (19%), and cattle (17%). Deer are abundant in the settlement in all periods and are represented by approximately equal numbers of roe deer (*Capreolus capreolus*) and red deer (*Cervus elaphus*). Considering the large body size of red deer and the general abundance of deer remains at the site, venison likely represented a significant part of the diet of the İnönü cave inhabitants. Across the chronological scope from the Chalcolithic to the Iron Age, the representation of significant mammal taxa in İnönü Cave is relatively stable over time. Pigs are well represented in all periods, peaking at 50% of specimens identified to species in the Late Bronze Age level. Goats peak in the Early Bronze Age level (29%) but are never the dominant taxon. The frequency of cattle is also reasonably consistent. However, it decreased during the Iron Age, when pig herding and deer hunting were the dominant economic activities in the faunal assemblage. The fact that deer have become an essential wild resource over time suggests that they remain a continuously available resource in this forested environment. However, both deer taxa decline in the Late Bronze Age, suggesting that perhaps swine herding replaced hunting as an economic activity at that time [23]. (Fig 3) [26].

Bioinformatics analysis identified the presence of a wide range of taxa, including Cyanobacteria, the *Acidobacteriota phylum*, the *Actinobacteriota phylum,* Bacteroidota, Chloroflexi, Firmicutes, Myxococcota, Proteobacteria, Cyanobacteria, and Nitrospirota species (Table 2).

The *Acidobacteriota phylum*, also called Acidobacteria, encompasses a group of bacteria widespread in diverse environments, including soils, freshwater, and marine sediments. This phylum ranks among the most abundant bacterial groups in soil and plays a crucial role in nutrient cycling and the degradation of organic matter. Acidobacteria exhibit a range of metabolic capabilities and are recognized for their involvement in various biogeochemical processes, including nitrogen and carbon cycling [28–30]. Despite their ecological importance, the metabolic and physiological properties of Acidobacteria are not fully known, nor have most of their genomes been described. Recently, efforts have been made to sequence and annotate the genomes of various Acidobacteria strains to improve our understanding of their biology and

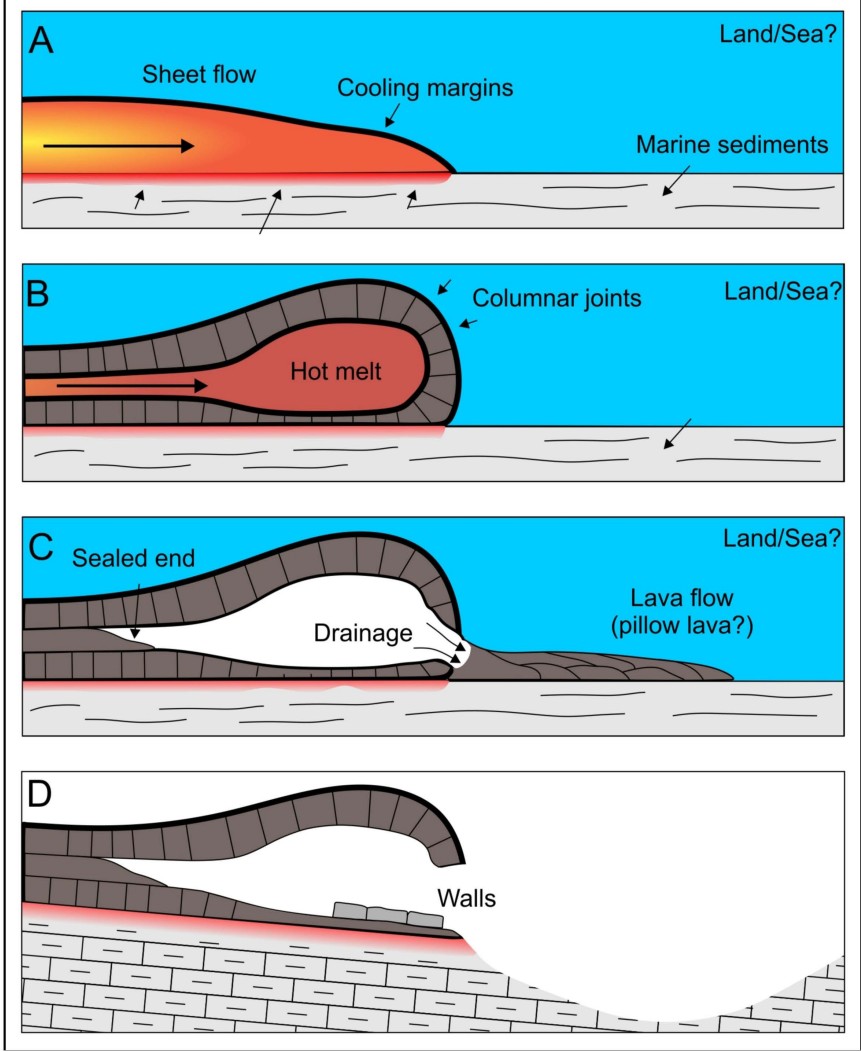

**Fig 1. The development of volcanic caves has four stages (adapted from Carracedo Sánchez et al., 2012).** In the first stage, sheet flow formed flow occurred. Then still hot melt supplied the cooled front and caused expansion of hot melt volume. After that, new branches of lava flow have been formed with the collapse of the side wall or an internal tube network, which may cause the melt drained and almost cooled lava flow sealed to the end of the tube. In the last stage, due to the uplift of the region, an erosional regime has started [25].

ecology [28,30]. Many Actinobacteria genomes have been sequenced, providing insight into their metabolic diversity and potential biotechnological applications [31–33]. Acteroidota (formerly Bacteroidetes) is a phylum of Gram-negative bacteria known for their ability to degrade complex organic compounds and their involvement in various ecological processes [34].

Chloroflexi is a phylum of Gram-negative bacteria commonly found in various environments, including soil, freshwater, and marine sediments. They are also found in anaerobic environments like hot springs and deep-sea hydrothermal vents. Chloroflexi bacteria are characterized by their green-colored chlorosomes, which contain bacteriochlorophyll pigments that enable them to harvest light energy for photosynthesis. They are also unique in possessing the enzyme chlorophyllide, an oxygenase required for the biosynthesis of chlorophylls and bacteriochlorophylls [30,35]. Firmucutes is a phylum of Gram-positive

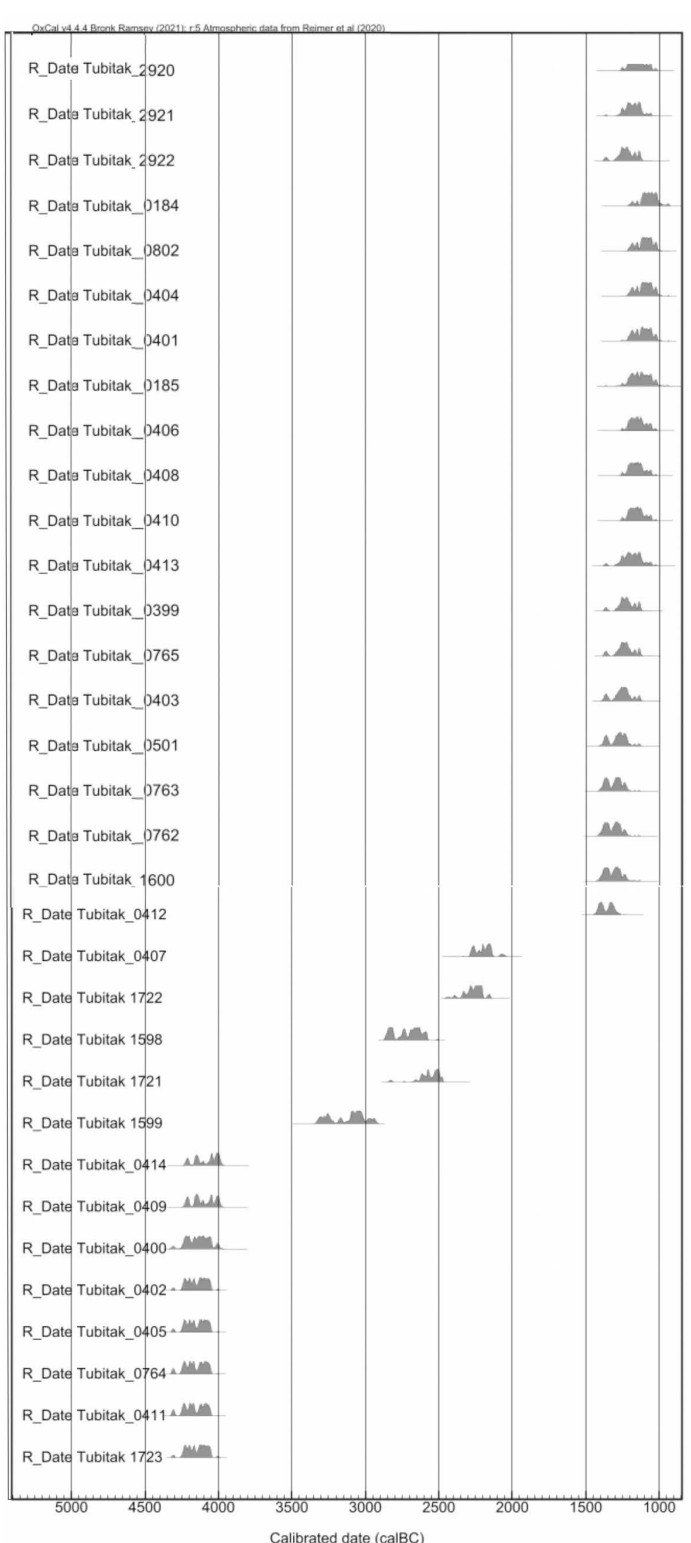

**Fig 2. Calibrated dates of İnönü Cave's all levels.** The 14C ages were calibrated using OxCal v4.4.4 (Bronk Ramsey, 2021), and the atmospheric data from the calibration curve of (Reimer et al. 2020).

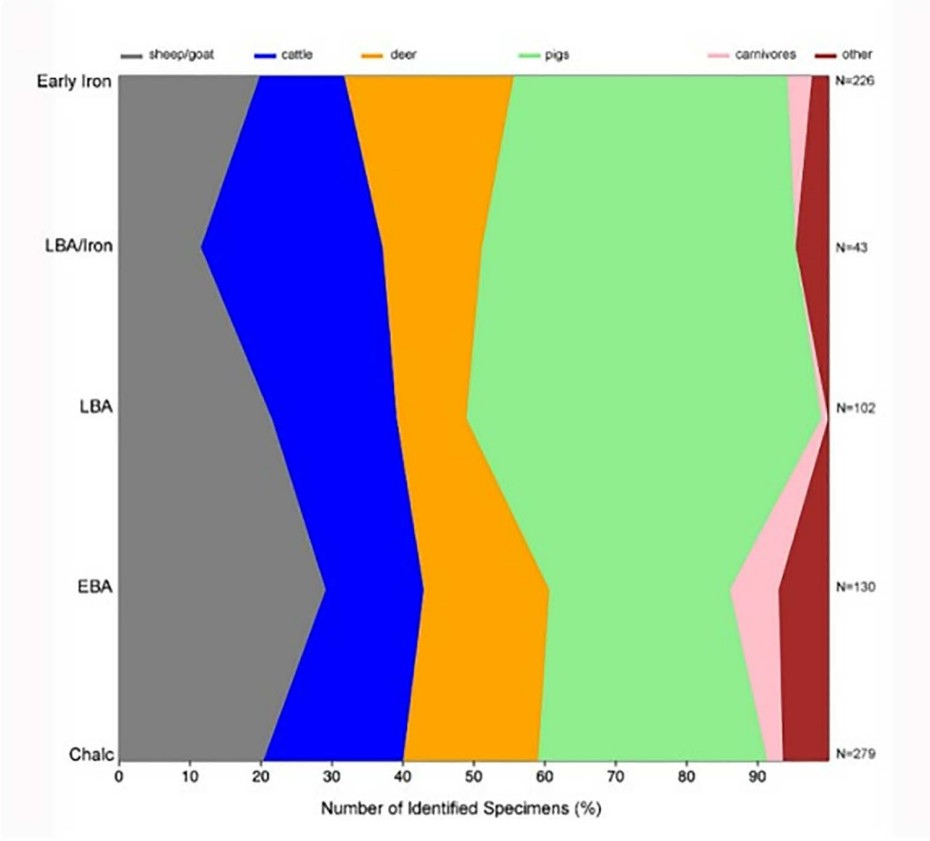

**Fig 3. Number of Identified Specimens (%) [26].**

**Table 1. Table showing the calibrated radiocarbon dates of the levels from which the soil was sampled and the trenches, altitude, and sample collection dates (© Archive of İnönü Cave Project).**

| Levels | Ages | Calibrated Dates | Number of Soil Samples | Trench of Soil Samples | Altitude of Soil Samples(meter) | Collection Date |
|--------|------|------------------|------------------------|------------------------|----------------------------------|-----------------|
| I | Medieval Age | – | – | – | – | – |
| II | Early Iron Age | 1231−979 Cal. BC. | 811 (A1) | İ/8 | 235,70 | 01.12.2022 |
| III | Late Bronze Age | 1436−1123 Cal. BC. | 812 (A2) | İ/8 | 235,20 | 01.12.2022 |
| IV | Early Bronze Age | 3126−2133 Cal. BC. | 813 (A3) | H/7 | 234,09 | 01.12.2022 |
| V | Chalcolithic Age | 4260−3976 Cal. BC. | 814 (A4) | H/7 | 233,78 | 01.12.2022 |

bacteria found in various environments, including soil, water, and the gastrointestinal tract of animals. They are known for their ability to form spores, which can resist harsh environmental conditions such as high temperatures, low moisture, and UV radiation. Firmicutes bacteria are characterized by thick cell walls consisting of peptidoglycan and other complex polysaccharides. They are also diverse in their metabolic activities, including fermentation, respiration, and sulfur reduction [36,37].

Myxococcota is a phylum of Gram-negative bacteria commonly found in soil environments. These bacteria are known for their unique life cycle, which involves the formation of multicellular fruiting bodies for reproduction. They can also move using a gliding mechanism and are known to prey on other bacteria [38–40].

**Table 2. Antibiotic resistance genes studied in soil samples.**

| Primer Name | Primer Sequence | Primer Sequence | Gene Region | Target | Bp | Tm |
|---|---|---|---|---|---|---|
| 22-109 | blaTEM-F | CATTTCCGTGTCGCCCTTATTCC | blaTEM | Antibiotic resistance | 828 | 61 |
| 22-110 | blaTEM-R | GGCACCTATCTCAGCGATCTGTCTA | blaTEM | Antibiotic resistance | 828 | 61 |
| 22-111 | tetA-F | AGGTGGATGAGGAACGTCAG | 16SrRNA | Antibiotic resistance | 63,5 | |
| 22-112 | tetA-R | AGATCGCCGTGAAGAGGCG | 16SrRNA | Antibiotic resistance | 63,5 | |
| 22-113 | intl1-F | GATCGGTCGAATGCGTGT | 16SrRNA | Antibiotic resistance | 196 | 60 |
| 22-114 | intl1-R | GCCTTGATGTTACCCGAGAG | 16SrRNA | Antibiotic resistance | 196 | 60 |
| 22-115 | OXA58-F | GCAATTGCCTTTTAAACCTGA | 16SrRNA | Antibiotic resistance | 152 | 63 |
| 22-116 | OXA58-R | CTGCCTTTTCAACAAAACCC | 16SrRNA | Antibiotic resistance | 152 | 63 |

Proteobacteria is a large and diverse phylum of gram-negative bacteria that includes a wide range of species with various metabolic capabilities, lifestyles, and ecological roles. They are found in various environments, including soil, water, and hosts, and play essential roles in nutrient cycling, biodegradation, and pathogenesis [41,42]. Proteobacteria are classified into six classes: Alpha-, Beta-, Gamma-, Delta-, Epsilon-, and Zetaproteobacteria, based on their phylogenetic relationships and distinctive characteristics. Alphaproteobacteria are commonly found in symbiotic associations with eukaryotes, while Betaproteobacteria often utilize alternative electron acceptors and can degrade complex organic compounds [43].

Cyanobacteria, also known as blue-green algae, are a phylum of photosynthetic bacteria widely distributed in diverse habitats, ranging from aquatic to terrestrial environments. They are characterized by their ability to perform oxygenic photosynthesis, which makes them critical primary producers in many ecosystems. Cyanobacteria are also known for their ability to form symbiotic associations with various organisms, such as fungi, plants, and animals [44,45]. The phylum Cyanobacteria comprises many morphologies, including unicellular, filamentous, and colonial forms. The pigments they use for photosynthesis are chlorophyll and phycobiliproteins, which give them their characteristic blue-green color.

Nitrospirota is a phylum of bacteria known for its role in nitrogen cycling. They are spiral-shaped and motile, and their metabolism is primarily anaerobic. Nitrospirota are found in various environments, including soils, sediments, and wastewater treatment systems. They are known for their ability to oxidize nitrite and nitric oxide to nitrate, as well as their ability to reduce nitrate to nitrite or nitrogen gas. Nitrospirota has also been implicated in denitrification, a process that converts nitrate to nitrogen gas, which can help mitigate the effects of nitrogen pollution in aquatic environments [46,47].

## Materials and methods

The excavations of İnönü Cave are carried out under the license granted by the Ministry of Culture and Tourism of the Republic of Turkiye to one of the authors, Hamza Ekmen. All necessary permits were obtained for the described study, which complied with all relevant regulations. The license granted covers all scientific studies related to İnönü Cave.

All primary archaeological artifacts are stored at the Zonguldak Provincial Karadeniz Ereğli Archaeological Museum, and secondary artifacts that require laboratory work are stored at the Zonguldak Bülent Ecevit University Archaeology Department Laboratory by the relevant license.

The soil samples from the levels briefly summarised above were taken from the defined fills of each level under sterile conditions and in the presence of archaeologists from the İnönü Cave Project Team (Figs 4 and 5).

The fillings of the four archaeological levels were verified with C14 carbon analyses and evaluated by Assoc. Prof. Dr. Hamza Ekmen and Assoc. Prof. Dr. F. Gülden Ekmen.

Soil samples were taken from these layers under the control of the excavation head Assoc. Prof. Dr. Hamza Ekmen and the responsible researcher Assoc. Prof. Dr. Gülden Ekmen, using materials prepared by Beun Faculty of Pharmacy, Department of Pharmaceutical Microbiology, Head of the Department of Pharmaceutical Microbiology, Assoc. Prof. Dr.

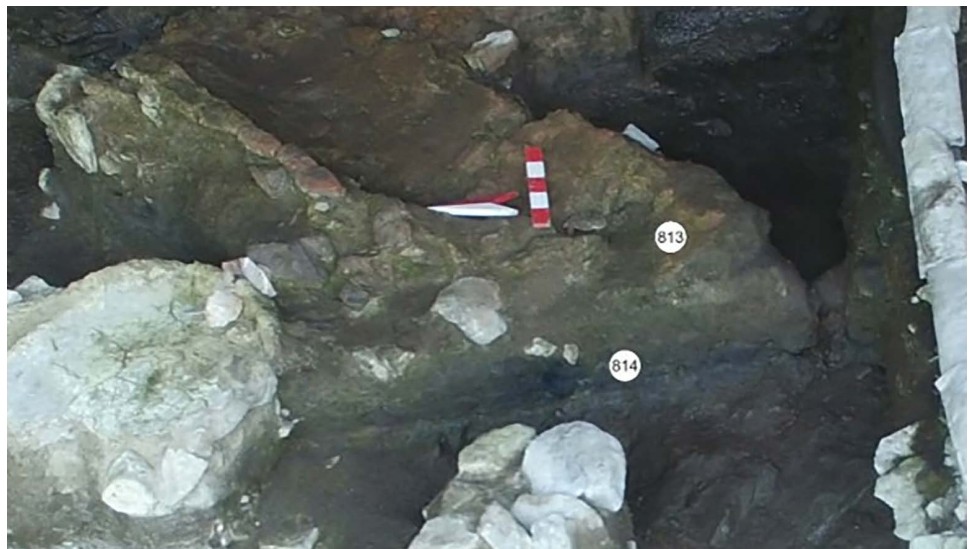

**Fig 4. Archive of Inonu Cave Project 813 (A3) and 814 (A4).**

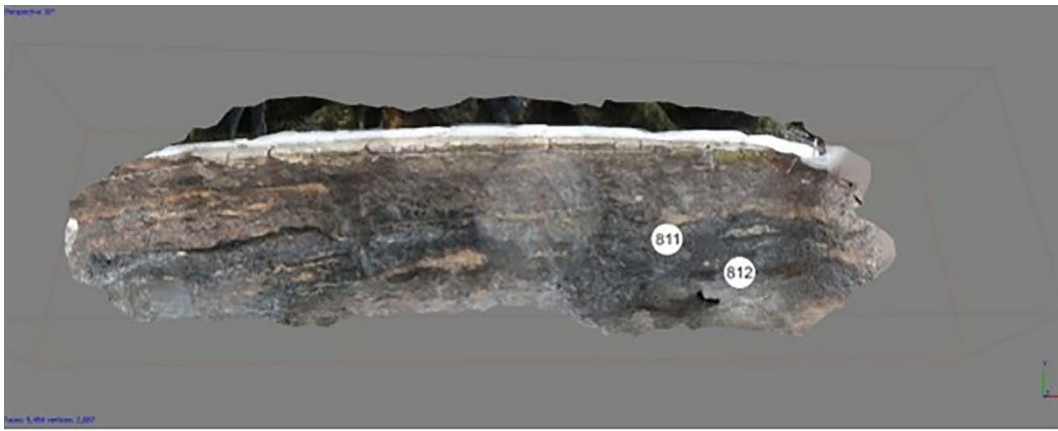

**Fig 5. Archive of Inonu Cave Project 811 (A1) and 812 (A2).**

Sukran Ozturk, sterilized in an autoclave (Nüve) at 121°C for 15 minutes in the Pharmaceutical Microbiology Laboratory of Beun Faculty of Pharmacy, using antisepsis conditions (Eppendorfs, falcon tubes, steel spatulas, steel collecting spoons) and transferred to the laboratory environment under sterile conditions (Figs 3 and 4). Detailed information is shown in Table 1. Soil samples were stored at −80°C for analysis.

Systematic excavations have been carried out in the cave since 2017, and five cultural levels have been uncovered. Analogical analyses of the archaeological material found in these levels were confirmed by the dates determined by calibrated radiocarbon analyses on 30 samples (Fig 2).

Level I was inhabited during the Middle Ages [40]. Level II (Sample 811), dated to the Early Iron Age, yielded simple stone architectural remains and a votive pit adjacent to the cave wall. A large number of sherds of Barbarian ware/Coarse ware and Handmade Glossy ware (Knobbed ware/Buckelkeramik) found in the votive pit indicates that the people of this

level were connected with the Iron Age cultures of Thrace, South Marmara, and Aegean coasts. Based on the large number of spindle whorls found on this level and in the votive pit, it can be concluded that the leading subsistence economies of the inhabitants of this level were animal husbandry and textile production. It is possible to relate the small finds from this level to the people who entered Anatolia with the Aegean/Balkan migrations at the end of the Late Bronze Age [26,27].

Level III was inhabited during the Late Bronze Age (sample 812). The wooden floors of four large and two small buildings were unearthed during the excavations [48]. The inhabitants of Level III built these floors to create a flat and dry living space on the sloping and humid cave floor. Most artifacts recovered from the floors are made of Bronze and consist of tools, weapons, and ornaments. Some of these findings are similar to those found in the cities of the Hittites, who established a central authority in Anatolia in the second millennium BC. This raises the question of whether the communities living in this cave were related to the Kashka or Pala-Tummana communities, which are frequently mentioned in Hittite texts and known to have lived in the north of the Hittite core5 [27].

Level IV was identified in a limited area and dated to the Early Bronze Age (sample 813). The finds recovered show some traces of simple production, probably at the household level. Vessels used to obtain secondary products from milk, evidence of pottery production, animal bone cut for tool making, burnt stone blades, and remains of stone tool production reflect the local production activities within the cave. Based on these data, it is suggested that İnönü Cave was a workshop area where small-scale production of a range of products and commodities was carried out at certain times of the year during the Early Bronze Age [49].

Level V, which bears traces of the earliest inhabitants of the settlement, was occupied during the Chalcolithic Age (sample 814). Analogical analyses of the finds produced with various raw materials such as gold, copper, agate, spondyl shell, bone, and flint at this level reveal that İnönü Cave is connected with the cultures in the Eastern Balkans, especially the coastal cultures. This underlines the maritime cultural influences between the two regions and provides clues about trade-related social interaction and common technological development [50].

## Soil DNA sampling and extraction

**aDNA extraction.** Before DNA extraction, all lab surfaces were sterilized using 50% bleach and exposed to UV light for 15 minutes. Pipettors, tubes, and racks were autoclaved. Filtered pipette tips were used throughout the isolations, gloves, centrifuge rotors, and the laboratory oven (for the incubation step) were sterilized regularly. All stages were carried out in a clean room and a separate cabin.

DNA isolation of ancient DNA Soil samples (1100–1200 BC Iron Age; 1200–1360 BC Late Bronze; 2000–3000–3500 B.C. Early Bronze; 4300 BC Chalcolithic) was performed with the "MN NUCLEOSPİN Soil DNA Extraction Kit." (The MN NUCLEOSPİN Soil DNA Extraction Kit was used to isolate aDNA from the soil samples.) Each sample was analyzed separately using both Buffer SL1 and Buffer SL2

MN NUCLEOSPIN Soil DNA Extraction Kit was selected due to its established reliability in extracting ancient DNA (aDNA) from challenging matrices of soils that frequently have high concentrations of PCR inhibitors, including humic acids. The innovative technology included in the kit for the removal inhibition guarantees higher purity and yield of the DNA, which is essential for downstream applications in research on ancient microbiomes. Additionally, the dual-buffer system (SL1 and SL2) enhanced the lysis efficiency for a wide range of soils, thus increasing DNA extraction from Gram-positive and Gram-negative bacterial communities [12].

**PCR and bioinformatics.** Small 16S gene region primers (16SV3-F/16SV3-R) were used for the qPCR test to see if PCR inhibition happened. The inhibition test results showed that inhibitors needed to be removed. SL1 buffer DNA isolates were diluted 1:10 to do this, and GoTaq® G2 Flexi DNA Polymerase was used for PCR.

Primers 16SV3-F and 16SV3-R were chosen because they specifically target the hypervariable V3 region of the 16S rRNA gene, allowing for high-resolution taxonomic assignment in bacterial communities. The same region is well-reported in ancient DNA research due to its ability to produce robust phylogenetic information even when dealing with

sheared DNA [20,29]. To detect antibiotic resistance genes, blaTEM-, tetA-, intl1-, and OXA58-specific primer pairs were selected due to their extensive validation in environmental and ancient microbiology [51,52]. These primers were shown to be more specific and sensitive in detecting resistance genes from degraded DNA templates, giving a true reflection of ancient resistomes [53]. The 16SV3-F/16SV3-R primers were selected for their specificity in targeting the hypervariable V3 region of the 16S rRNA gene, providing high-resolution taxonomic identification of bacterial communities. This region is well-documented in ancient DNA studies because it can generate reliable phylogenetic data even from fragmented DNA. Primer pairs targeting blaTEM, tetA, intl1, and OXA58 were chosen for antibiotic resistance gene detection based on their widespread validation in environmental and archaeological microbiology. These primers have demonstrated high specificity and sensitivity in detecting resistance genes from degraded DNA templates, ensuring an accurate representation of ancient resistors. For the target of antibiotic resistance genes blaTEM-F/blaTEM-R [48], tetA-F/tetA-R [45], intl1-F/intl1-R [52], OXA58-F/OXA58-R [54], primer pairs and universal Illumina adaptors added primers 16SV3-F/16SV3-R for microbiota analysis were used. The PCR reaction profile includes the following steps: 94°C for 2 min (initial denaturation), 94°C for 1 min (denaturation), 63°C for 45 sec (annealing), 72°C for 1 min (extension), 72°C for 5 min (final extension). The PCR reaction mixture includes PCR buffer 5 µl, forward primer 1.25 µl, reverse primer 1.25 µl, dNTP 2 µl, MgCl2 2.5 µl, polymerase 0.125 µl, PCR grade water 11.135 µl, and template DNA 1.5 µl, with a total reaction volume of 25 µl.

PCR products were quantified with the Qubit Fluorometer using the dsDNA H.S. Assay Kit (Thermo Fisher Scientific, Waltham, MA, USA). The 16S Metagenomic Sequencing Library Preparation Kit was used and purified with the AMPure purification kit. Library fragment sizes were checked with the Agilent Technologies 2100 Bioanalyzer, then sequenced with Illumina iSeq (Illumina, San Diego, CA, USA) (2x150 P.E.).Illumina iSeq (2x150 paired-end) sequencing was selected as it can handle a large amount of data rapidly and is highly accurate for short reads, which is crucial when working with ancient DNA. The platform can generate large volumes of high-quality data cost-effectively, making it ideal for in-depth metagenomic research.

Furthermore, the 150 bp paired-end reads are suitable for amplifying the short, fragmented ancient DNA pieces, allowing for optimal recovery of informative sequences [55]. Illumina iSeq (2x150 paired-end) sequencing was selected due to its high-throughput capabilities and exceptional accuracy in short-read sequencing, which is essential for ancient DNA studies. The platform's ability to produce large volumes of high-quality data at a lower cost makes it ideal for comprehensive metagenomic analyses. The 150 bp paired-end reads are particularly suited for amplifying ancient DNA's short, fragmented nature, ensuring maximal recovery of informative sequences. OBITOOLS was used because it has specific expertise in metabarcoding analyses. It offers a complete range of tools to filter sequences, assign taxonomies, and correct errors. These tools are of utmost importance when dealing with ancient DNA datasets that will degrade and get contaminated [28]. The SILVA database was employed to identify species since it contains a vast and structured inventory of ribosomal RNA sequences. The inventory provides trustworthy taxonomic information that is essential in the accurate description of ancient microbial communities [30]. Quality control was enforced strictly through the utilization of FastQC, based on a stringent Phred score threshold of 30, to confirm the accuracy of the sequence data, reducing errors and problems ubiquitous to ancient DNA sequencing [55]. OBITOOLS was employed due to its specialized capabilities in metabarcoding analyses. It offers a comprehensive suite of tools for sequence filtering, taxonomic assignment, and error correction, which are critical for handling ancient DNA datasets prone to degradation and contamination. The SILVA database was utilized for taxonomic identification because of its extensive, curated repository of ribosomal RNA sequences, providing high-confidence taxonomic resolution essential for the accurate characterization of ancient microbial communities. Quality control was rigorously enforced using FastQC, with a stringent Phred score threshold of 30, to ensure the reliability of sequence data, minimizing errors and artifacts common in ancient DNA sequencing. The low-quality ends of the sequences were filtered with Illumina paired-end code, determining a minimum threshold of 30 Phred scores [44,55]. Sequences were merged with the Illumina paired-end tool under default overlap parameters. Reads containing ambiguous bases or lengths less than 150 base pairs were filtered out to maintain data integrity. Chimeric sequences were detected

and filtered out using the uchime-denovo algorithm in VSEARCH. Taxonomic assignments were made with an 80% confidence threshold against the SILVA 138 database.

Rarefaction analysis was conducted to a depth of 90,000 reads per sample to normalize the sequencing effort for all samples. Unaligned sequences with the obigrep code were removed from the data. Adaptor primers were trimmed, and each was assigned to different files according to sample numbers with filter commands and the workflow of the OBITools program. Using the bionic and obi annotate commands selected in accordance with the workflow of the OBITools program, the repetitive sequences were cleaned, and the read numbers were added to the sequence information. Using Linux bash commands, sequences higher than one read and sequence lengths between 150–250 bp (16SV3-F/16SV3-R) were selected. The Obiclean command of the OBITools program was used to find and add to the data set sequences that change because of sequencing and PCR errors or real nucleotide variation in populations. The resulting filtered sequences were matched with the SILVAngs database, and genus-level results were obtained from the database (Table 2).

For 16S high-throughput sequencing, the forward and reverse reads were lined up with the 16SV3-F/16SV3-R primer pair. This produced a total of 100,000 reads for each sample. This made the 16S gene region bigger. After cleaning the raw data, approximately 90,000 sequence reads remained and were blasted using the SILVAngs database.

## Statistical analysis

We applied a series of statistical tests to assess the diversity and composition of microbial communities included in various archaeological levels. Alpha diversity metrics such as Shannon and Simpson indices were computed utilizing QIIME2 to compare species richness and evenness among the samples. Beta diversity was examined through Bray-Curtis dissimilarity and then plotted using Principal Coordinates Analysis (PCoA) to facilitate comparison of the microbial community structure among samples of various temporal groups. Differential abundance analysis was performed with the DESeq2 package in R to detect the significant differences in taxa abundance across the various archaeological levels. The P-values were adjusted for multiple testing using the Benjamini-Hochberg method with a significance threshold at $p < 0.05$. To detect antibiotic resistance genes, qPCR data were analyzed using standard curve methods to quantify the gene copy numbers. Statistical significance for changes in the abundance of resistance genes across various periods was determined using one-way ANOVA, followed by post hoc Tukey's tests where necessary.

## Results

The bioinformatic analyses revealed the characteristics of the various taxa and bacteria identified, including Cyanobacteria, Acidobacteriota phylum, Actinobacteriota phylum, Bacteroidota, Chloroflexi, Firmicutes, Myxococcota, Proteobacteria, Cyanobacteria, and Nitrospirota species. The introduction section mentions the characteristics of the various taxa and bacteria identified (Fig 6).

Bioinformatics analysis revealed a diverse microbial composition in soil samples from different archaeological levels of İnönü Cave. Taxonomic classification was conducted using SILVA 138.1 database with an 80% confidence threshold. The results indicated the dominance of Acidobacteriota (23–32%), Actinobacteriota (14–22%), Bacteroidota (5–12%), Chloroflexi (7–15%), Cyanobacteria (10–18%), Firmicutes (8–19%), Myxococcota (3–9%), and Proteobacteria (18–27%) across different periods. A comprehensive breakdown of taxa abundances is provided in Table 3. The variations in community composition suggest a strong correlation between microbial diversity and human activities over time.

Actinobacteria are known for their ability to produce a wide range of bioactive compounds, such as antibiotics, antifungal drugs, and antitumor agents. This has led to extensive research into their potential applications in medicine and biotechnology [56].

Another bacterial taxon that has been identified is Bacteroidota. They are commonly found in aquatic environments and soil. They are important in degrading complex organic matter such as polysaccharides and play a vital role in the carbon cycle [34]. Chloroflexi (specimen 812), which is more dominant than other species in the Late Bronze Age fill of the cave,

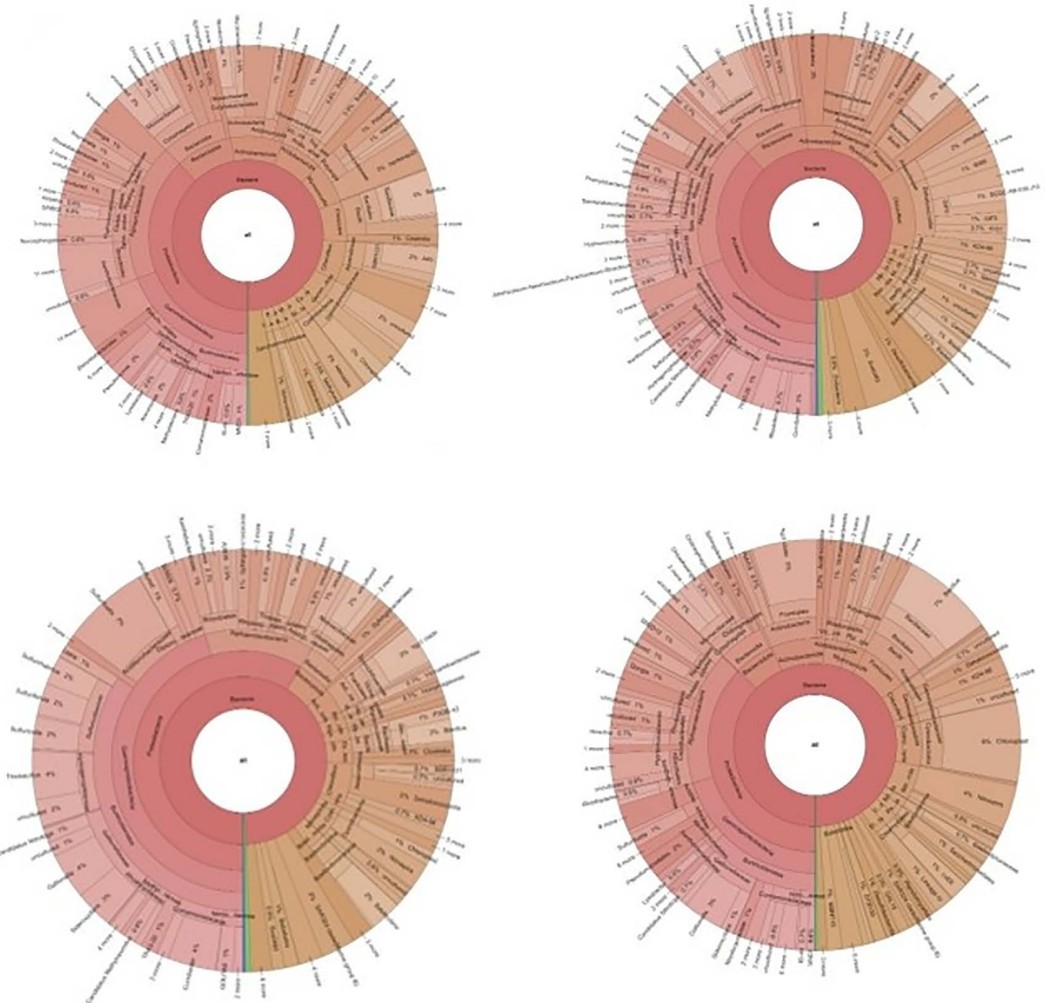

**Fig 6. The Krona charts depict the proportional distribution and taxonomic makeup of bacterial communities in soil samples obtained from four different levels: the Chalcolithic Age (6d) (sample 814), Early Bronze Age (6c) (sample 813), Late Bronze Age (6b) (sample 812), and Early Iron Age (6a) (sample 811).** The charts offer a hierarchical representation of bacterial diversity, enabling a detailed comparison of microbial populations throughout various historical periods. The investigation was conducted utilizing high-throughput sequencing of 16S rRNA genes, and taxonomy designations were determined using the SILVA database. These maps provide valuable information about the changes in microbial communities that historical human activities, environmental circumstances, and the passage of time have influenced. The outer rings depict lower taxonomic levels, such as family, genus, and species, while the inner rings depict higher taxonomic levels, such as phylum, class, and order. This arrangement offers a comprehensive perspective on microbial diversity. The results emphasize the notable occurrence of several bacterial phyla, such as Proteobacteria, Acidobacteriota, Actinobacteriota, Bacteroidota, Chloroflexi, Cyanobacteria, Firmicutes, and Myxococcota, and their variations over time.

is known for its ability to perform various metabolic activities, including photosynthesis, fermentation, and sulfur oxidation, and its density can be explained by the presence of groundwater in this layer. It is known that Bacteroidota, which was also detected in samples taken from the 812 fillings, is rare among bacterial communities and that the Bacteroidetes phylum, which is found at an average rate of 5% in the soil, is an essential anaerobic member of the output microbiome [57].

Firmicutes, most commonly found in the Chalcolithic Age (sample 814), is another phylum found in various environments, including soil, water, and the gastrointestinal tract of animals. Some important Firmicutes bacteria include the

**Table 3. Taxonomic Composition of Major Bacterial Phyla Across Archaeological Levels.**

| Phylum | Chalcolithic (%) | Early Bronze (%) | Late Bronze (%) | Early Iron (%) |
|---|---|---|---|---|
| Acidobacteriota | 23 | 28 | 30 | 32 |
| Actinobacteria | 14 | 17 | 20 | 22 |
| Bacteroidota | 5 | 8 | 10 | 12 |
| Chloroflexi | 7 | 9 | 13 | 15 |
| Cyanobacteria | 10 | 14 | 16 | 18 |
| Firmicutes | 8 | 12 | 16 | 19 |
| Myxococcota | 3 | 5 | 7 | 9 |
| Proteobacteria | 18 | 22 | 25 | 27 |

genera Bacillus, Clostridium, Lactobacillus, and Staphylococcus. Lactobacillus species produce fermented foods such as yogurt and cheese [36,37]. Bacillus species are known for producing antibiotics and enzymes, while Clostridium species are essential in producing biogas and other industrial products. Bioinformatics analysis results of the study conducted on İnönü Cave soil samples revealed decreases or increases in some bacterial communities at different ages. Accordingly, the first result is the decrease in the dominance of Myxococcota in the Early Iron Age (Sample 811) compared to the other three periods. Myxococcota has been extensively studied for its biotechnological potential, particularly for its production of secondary metabolites with antimicrobial, anticancer, and immunosuppressive properties. In addition, it has been investigated for its potential in bioremediation due to its ability to degrade various organic compounds [38,58].

It was determined that Proteobacteria increased in the early Bronze Age. The results were evaluated based on different information about the species of Proteobacteria. The Gammaproteobacteria are diverse and include many important human and animal pathogens and some photoautotrophic species. The Deltaproteobacteria are involved in sulfur and metal cycling, while the Epsilonproteobacteria include many pathogens that live in the digestive tracts of animals. Zetaproteobacteria are recently discovered bacteria involved in iron oxidation [41,59]. Proteobacteria have been extensively studied and are of significant interest in many fields, including microbiology, ecology, biotechnology, and medicine. They have been shown to produce various bioactive compounds, such as antibiotics, enzymes, and pigments, and have been utilized for bioremediation and biocatalysis. In addition, some species of Proteobacteria have been developed as model organisms for research in molecular biology and genetics [41,60]. Cyanobacteria (specimen 814), also known as blue-green algae and identified as dominant in the Chalcolithic level, are a phylum of photosynthetic bacteria widely distributed in various habitats from aquatic to terrestrial environments. They are characterized by their ability to perform oxygenic photosynthesis, making them critical primary producers in many ecosystems. Cyanobacteria are also known for their ability to form symbiotic relationships with various organisms, such as fungi, plants, and animals, and to fix atmospheric nitrogen, an important nitrogen source for many ecosystems [44,45]. Cyanobacteria have a long evolutionary history and are thought to have played a vital role in the evolution of the Earth's atmosphere. They are also of great ecological and economic importance as they are used in various biotechnological applications such as wastewater treatment, biofuel production, and food production [45].

In addition to identifying taxonomic diversity in the soil samples, this study explored evidence for antibiotic resistance genes in ancient sediments. The results of a study analyzing the antibiotic resistance genes in the İnönü Cave archaeological samples from different time periods were obtained by running the samples on a 2% gel. We detected the presence of different antibiotic-resistance genes in the samples from different ages. Specifically, we detected the tetracycline antibiotic resistance gene tetA in the sample from 4300 BC. Chalcolithic Age (814), the class 1 integron (Intl) antibiotic resistance gene in the samples from the 2000 - 3000 - 3500 BC. Early Bronze Age (813), and the oxacillinase gene in the samples from 2000–3000–3500 BC. Early Bronze Age. In sample 812 from the Late Bronze Age, OXA 58 was detected.

blaCtxm and qnrS antibiotic resistance genes were not found in the samples studied. These results are important as they are the first to show that antibiotic-resistant genes were present in the cave's sediments in prehistoric and early historical periods. These results suggest that antibiotic-resistance genes were present in bacterial populations during these ancient time periods, which has important implications for understanding the evolution and spread of antibiotic resistance over time. As detailed above, the results of the bioinformatics analysis show that the identified species represent nine separate phyla, and their dominance varies according to time periods. Among the diachronic changes we noted was a decrease in the dominance of Myxococcota in the Iron Age (sample 811) compared to the other three periods. In the Early Bronze Age (sample 813), there was a noticeable increase in the number of Proteobacteria, which put significant pressure on the microbiota. It was determined that the presence of cyanobacteria also increased in the Chalcolithic Age (sample 814).

The agarose gel image in Fig 7 displays the presence of antibiotic resistance genes in soil samples from different archaeological levels of İnönü Cave. Three genes (tetA, intl1, OXA58) were positive in three distinct samples, 814, 813, and 812, respectively (Fig 7).

Each positive sample lane displays a distinct band corresponding to the expected size of the amplified PCR product for the respective antibiotic resistance gene. The negative control confirms the absence of contamination (except for a slight band in blaTEM, which is found to be negative). The detection of these genes across different archaeological levels suggests the historical presence and persistence of antibiotic resistance in the soil microbiome of İnönü Cave.

The taxonomic fingerprint of microbial communities at the phylum level in soil samples collected from four separate cultural strata in İnönü Cave is depicted in this image (Fig 8). The analysis was conducted using the SILVA database. The taxonomic composition is determined by the proportional representation of different bacterial phyla in the samples, indicating microbial communities' variety and spread.

A1_S25_L001.clean: Soil sample obtained from the Early Iron Age.

A2_S26_L001.clean: Soil sample obtained from the Late Bronze Age.

A3_S27_L001.clean: Soil sample obtained from the Early Bronze Age.

A4_S28_L001.clean: Soil sample obtained from the Chalcolithic Age.

The bar charts depict the proportional distribution of bacterial phyla in each sample. Some of the noteworthy phyla are Proteobacteria, Actinobacteria, Acidobacteriota, Bacteroidota, Cyanobacteria, Firmicutes, and Myxococcota, among others. The chart uses different colors to represent each phylum, visually displaying the microbial variety at each archeological level. The taxonomic fingerprint reveals the variations in the makeup of bacterial communities throughout time, indicating the impact of past human actions and environmental changes on soil microorganisms (Fig 8).

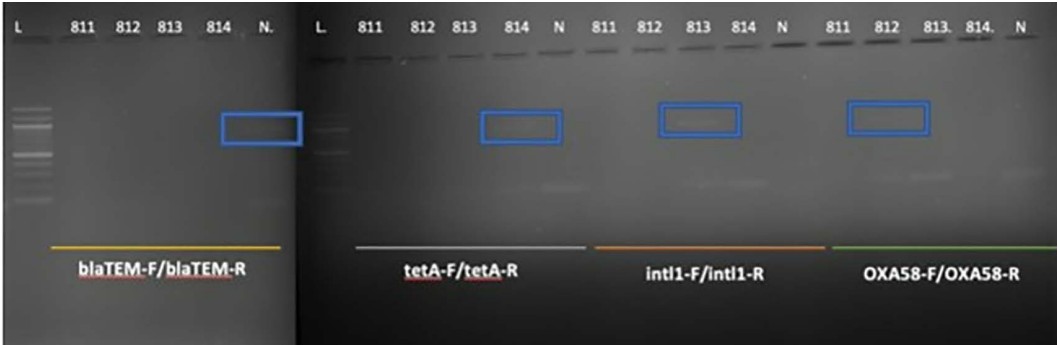

**Fig 7. Agarose gel electrophoresis image of PCR results for antibiotic resistance genes in soil samples from Inonu Cave.** Agarose gel electrophoresis image of PCR results for antibiotic resistance genes in İnönü Cave soil samples given in Fig 7. Raw data for this study can be seen Fig S1, Fig S2 and Fig S3 (in S1 File).

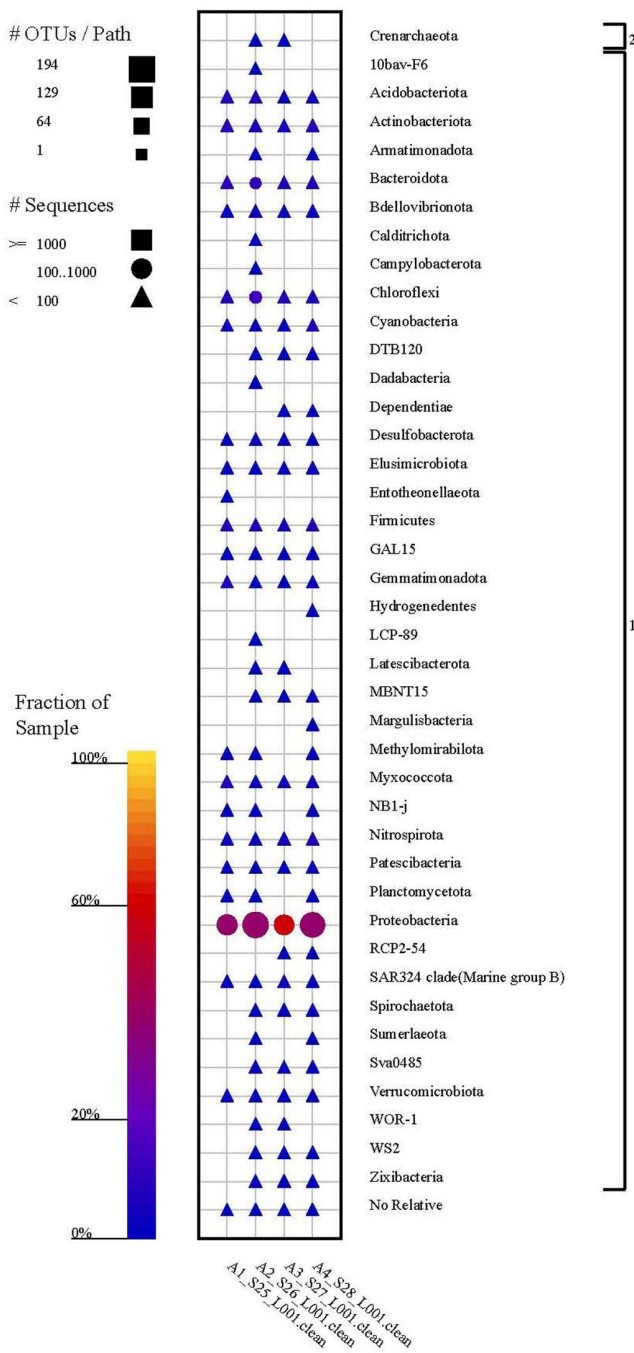

**Fig 8. The taxonomic composition of microbial communities in İnönü Cave soil samples was analyzed at the phylum level across several archaeological levels.**

The diagram highlights the ever-changing characteristics of microbial communities and their ability to adjust to different environments during the many time periods depicted in the İnönü Cave.

While significant microbial abundance and diversity were found in soil samples from four archaeological levels, our study does not reveal differences in microbial community structure and diversity. These differences may be related to

various consumer, disease, and nutritional factors. These microarchaeological findings and microbial characteristics may be attributed to different microbial responses to specific consumer habits. Hence, different consumption habits, patterns, and disease profiles lead to different compositions of the generated wastes and thus leave different microbial traces.

## Discussion

Soil is the most critical yet undiscovered reservoir of antibiotic resistance (A.R.), as it provides a biological summary of life from when it was found. The highly diverse and abundant soil microbiome may be a source for numerous genetic mechanisms with potential A.R. capacities. Paleoanthropogenic activities may transfer ARBs (Antibiotic Resistant Bacteria) and ARGs (Antibiotic Resistant Genes) to soil and indicate a selective dominance that could increase the abundance of natural ARGs. It is known that the soil's physical qualities significantly impact the activity and stability of antibiotics and ARGs in the soil matrix [35]. Additionally, extracellular DNA harboring ARGs may remain in soil matrices [61,62].

Studies on microbiota and genome analyses on soil samples obtained from archaeological deposits have gained importance in recent years. These studies have revealed the factors behind devastating historical epidemics, identified unknown characteristics of prehistoric pathogens, and provided more comprehensive information about humans' ancestral microbiota [1,12].

In this study, microbial communities and antibiotic resistance genes were identified by taking intact samples using sterile methods from the defined sediments of the İnönü Cave, numbered A1 (sample 811), A2 (sample 812), A3 (sample 813), and A4 (sample 814). By detecting the soil microbiome and antibiotic resistance genes in the soil samples in question, the effects of the lifestyles, consumption habits, disease profiles, and/or treatment options of the İnönü Cave communities on the microbiota were investigated. The detection of microbial DNA in soil samples taken from the cave has made it possible to understand the evolution of microorganisms and their symbiotic relationship with the human population. Revealing possible diseases of the relevant eras, antibiotic resistance gene profiles from past to present, and the treatment options applied in the region during those periods have also provided us with information.

The presence of ARGs such as tetA, intl1, and OXA58 in soil samples from İnönü Cave, spanning various archeological levels, offers convincing proof of these genes' early origins and enduring existence. Detecting tetA in the Chalcolithic Age sample implies that mechanisms for antibiotic resistance existed well before the introduction of contemporary antibiotics. This discovery is consistent with the resistome hypothesis, which suggests that numerous ARGs come from soil bacteria and have lived in microbial communities for thousands of years.

The intl1 gene, frequently present in clinical isolates and linked to Proteobacteria, was identified in the sample from the Early Bronze Age. This particular integron is renowned for its capacity to acquire and distribute ARGs, emphasizing the possibility of ancient horizontal gene transfer occurrences. Intl1 indicates that ancient microbial communities were already involved in genetic exchanges that could improve their survival ability in different settings.

The OXA58 gene, an indicator of resistance to carbapenem antibiotics, was detected in the Late Bronze Age sample. Sulfur analogs affect the production of carbapenems [12]. The fact that İnönü Cave is a natural volcanic cave and the presence of sulfur-rich environments, such as natural water sources within it, suggests that environmental factors significantly impacted the resistance profiles of ancient microbial communities. These findings emphasize the complex nature of the evolution of antibiotic resistance, which is driven by both human actions and natural environmental conditions. Another result is that the number of Proteobacteria increased noticeably in the Early Bronze Age, which put significant pressure on the microbiota. In similar studies conducted within the same scope, the characteristics of Proteobacteria have been reported, and it has been determined that they are widespread and dominant [57,63,64]. Proteobacteria, which are frequently observed to lead in the natural structure of the soil, are of great importance in the global carbon and nitrogen cycle [42,59]. As feeding bacteria, it is expected that Proteobacteria, which uses complex organic matter to metabolize, was found to be dominant in our study. It has been reported that Proteobacteria are dominant in soil samples taken from an archaeological site within a coal basin in Germany, close to the coal seam [65].

The metabarcoding data demonstrated substantial changes in the makeup of microbial communities during several historical periods. The prevalence of Proteobacteria in the Early Bronze Age sample indicates a microbial reaction to environmental shifts or human practices, such as agriculture and animal husbandry that were widespread throughout this period. Proteobacteria are recognized for their involvement in the nutrient cycling process, namely in the carbon and nitrogen cycles. The higher prevalence of Proteobacteria could indicate an increase in the amount of organic matter added to the soil and an improvement in soil fertility during this time.

Conversely, the sample from the Chalcolithic Age exhibited a significant rise in Cyanobacteria, which can convert nitrogen and flourish in situations with abundant light and nutrients. This change suggests that the people living in İnönü Cave during the Chalcolithic period may have participated in activities that encouraged photosynthetic bacteria development. This could have had an impact on the fertility of the soil and the overall functioning of the ecosystem.

The decrease in the dominance of Myxococcota in the Early Iron Age level of the cave can be explained by the fact that the relevant level is the first cultural fill under the Medieval fill in the cave. Soil analyses performed on samples taken from an archaeological site in a coal basin, similar to that of İnönü Cave in Germany, also indicated a similar result [39]. Moreover, the fact that the Early Iron Age settlement in the cave presented a limited and even short-term settlement character may be the second reason explaining this situation. The decline in the prevalence of Myxococcota in the Early Iron Age sample, compared to other periods, indicates alterations in soil composition and microbial relationships. The reduced presence of Myxococcota may suggest changes in soil moisture or organic matter content, as these organisms are recognized for their predatory behavior and intricate life cycles.

Overall, the differences in microbial community composition among various archaeological levels offer valuable information on the ecological effects of past human actions and shifts in the environment. These discoveries enhance our comprehension of the interactions between ancient communities and their surroundings and the enduring impacts of their activities on soil microbiomes.

Chloroflexi (sample 812), which was more dominant than the other species in the Late Bronze Age fill of the cave, is known to be a common bacterium in various environments, including soil, fresh water, and marine sediments. The abundant fresh groundwater in the cave can explain the widespread presence of this bacterium in İnönü cave.

The phylum Acidobacteriota, also known as Acidobacteria, is more abundant in the Late Bronze Age fill (sample 812) than in other fills. This group of bacteria is common in various environments, such as soil, freshwater, and marine sediments. Therefore, the predominance of this bacteria in the cave should be considered a natural result.

Another group of bacteria that predominates in the same fill (sample 812), Bacteroidota (formerly known as Bacteroidetes), is generally found in aquatic environments, soil, and the intestines of animals, including humans. It is known that the Bacteroidetes phylum, which is reported to be rare among bacterial communities, with an average presence in the soil of 5%, is a crucial anaerobic member of the intestinal microbiome [57]. Considering that the leading subsistence economies of the Late Bronze Age inhabitants of the cave were animal husbandry, agriculture, and hunting, it is possible to explain the dominance of this bacterium by intense contact with manure.

Cyanobacteria, which are dominant in the Chalcolithic Age (sample 814) and bear the traces of the earliest inhabitants of the settlement, are known to be found in rocks such as chert, shale, or mud deposits exposed to calcium carbonate-based environments [66]. It is predicted that the cyanobacterial population layer may have originated from aquatic animals such as fish, frogs, or turtles consumed by the Chalcolithic Age people in addition to the natural environment. In addition, it is known that the Chalcolithic Age inhabitants plastered the sloping floor rock of the cave, which was wet due to groundwater, with clay and compacted it into a flat and relatively dry environment, which is thought to have created a suitable environment for this bacterial community. Firmicutes are most commonly found in the Chalcolithic Age (sample 814) bacteria. The density of firmicutes at this level can be explained by the fact that it depends on the natural structure of the soil, as well as the milk consumed and the secondary milk products (yogurt, cheese, and butter) produced within the scope of animal husbandry, which is the primary subsistence economy [36,37]. Another species detected in our study is the Actinobacteria phylum, also known

as Actinobacteria. It is emphasized that Actinobacteria are the most abundant group in raw soil, liquid, and undisturbed sludge [31–33]. The widespread occurrence of this bacterium in level V, the oldest level in the cave, can be explained by the fact that it is the level closest to groundwater, in other words, the level with the most mud.

In level V fill is Nitrospirota, a bacterial phylum known for its role in the nitrogen cycle. Nitrospirota: It is found in various environments, including soil, sediment, and wastewater treatment systems. This feature explains its frequent occurrence in the muddy fill at the bottom level [67].

Another goal of this study is the detection of A.R. genes. Antibiotic resistance is a global phenomenon with grave epidemiological consequences. Although the spread of antibiotic resistance has generally been associated with selection resulting from the clinical use of antibiotics, recent studies have shown that resistance genes drive the global proliferation of antibiotic resistance in fertilizers, biosolids application, drinking water, the food chain, wastewater, dyestuffs, antibiotic compounds that exert selective pressure, and natural sources. Studies have also demonstrated potential transfer to relevant bacteria via environmental reservoirs. All these findings showed that resistance genes are linked to anthropogenic factors. Although some evidence correlates anthropogenic factors with high levels of antibiotic resistance in soil, it is becoming increasingly clear that unaffected and undisturbed soils contain a wide variety and abundance of antibiotic-resistant bacteria. This has led to the resistome hypothesis, which proposes that many pathogen-associated antibiotic resistance genes originate from antibiotic-producing soil bacteria and reach pathogens via horizontal gene transfer [62].

Numerous studies of natural antibiotic resistance in soil show that soil contains high levels of natural ARB carrying various ARG sequences independent of anthropogenic activities. However, under some conditions, anthropogenic activities can increase soil A.R. The fact that ARGs were present in the environment long before the age of antibiotics strongly supports that they play an ecological role unrelated to clinical use [68]. Class 1 integrons (intl), mobile DNA elements that can capture and transport genes, are perhaps the most common vectors of ARGs and are, therefore, frequently used as markers for source tracing of A.R. propagation in the environment.

In the Inönü cave samples, tetA was detected in the Chalcolithic Age fill (sample 814), intl1 in the Early Bronze Age fill (sample 813), and the OXA58 AR gene was detected in the Late Bronze Age fill (sample 812).

It is known that the TetA gene is found in places where livestock, plants, and vegetables are concentrated. Because organic soil fertilizers such as animal manure and biogas digestate often contain bacteria-carrying resistance genes (R.G.s) against antimicrobial substances and mobile genetic elements (MGEs). Moreover, although soil-amended antibiotics such as tetracycline and tylosin are tightly adsorbed to clay particles, they are still biologically active and, therefore, can form ARB. It has been reported that fertilizer increases the amount of Bacteroidetes and Firmicutes species in the soil and accordingly creates sul2, tet(W), and intl2 resistance genes [46]. In the Chalcolithic Age, which was the subcultural filling of caves, people's primary living economies depended on animal husbandry, agriculture, hunting, and gathering. Considering that the TetA gene was detected at this level, it should be considered a typical result.

Class 1 integrons (intl1), detected in Early Bronze Age fill, are the most common type of integrons found in clinical isolates, and Proteobacteria commonly found in freshwater and soil harbor these genes [47]. While the dominance of Proteobacteria in samples from the Early Bronze Age was noted in our study, it is clear that the intl1 resistance gene found may originate from this population. Intl1 resistance gene is also known as a resistance gene against cephalosporins. In this context, when we examine the structure of cephalosporins, we can see that they originate from a dye called Aniline [61]. Aniline, one of the coal tar derivatives and their combinations, is known in history as a compound used subcutaneously to reduce rheumatism, neuralgia, and temperature [69,70]. Today, modern medicine treats acute syndromes of poisonings and gastrointestinal diseases with activated charcoal therapy [62]. Considering that the excavation area subject to the study is a coal deposit area, the presence of the intl1 resistance gene can be explained by the aniline dye, that is, the coal used by human communities that were perhaps over-exposed to Proteobacteria in the relevant period, to get rid of the acute syndromes of gastrointestinal diseases. Interestingly, this possible treatment method was discovered by prehistoric human communities.

In the light of all this data, it is understood that human activities affect the soil. After the paleoanthropogenic effect disappears, organic matter loss occurs due to mineralization and transformation of microorganisms in the soil. However, despite all this, the ancient anthropogenic impact on soil can be preserved in soil microbiota and activities [18,57]. High organic matter input often stimulates microbial activity, leading to increased microbial biomass and enzyme activity. There is evidence that paleosols in the layers of prehistoric settlements or around archaeological monuments carry a record of the past's environmental conditions and the life of ancient people [1].

## Conclusion

When the few studies conducted jointly by microbiology and archeology and our study results were evaluated, it has been that while existing bacteria could be the source of the antibiotic-resistance genes found, some substances and dyes used for treatment or nutrition purposes also had the potential to create these resistance genes. It has been revealed through different studies that the vital activities of the human occupants of İnönü Cave include animal husbandry, agriculture, hunting, gathering, and weaving, and the bacterial populations and resistance genes detected in our study also supported this archaeological information. The antibiotic-resistance genes detected in this study reveal that humanity has used different methods and compounds to treat diseases since prehistoric times. In addition, it is a volcanic and water-rich cave, and some of the substances produced in this environment are sulfur, sulfur, etc. It plays a significant role in both the formation of bacterial populations and the presence of resistance genes. Another significant result is that antibiotic resistance genes have existed since prehistoric times and that the causative agent may be not only antibiotic use but also environmental factors, bacterial population and diversity, different trace elements in the soil, and dyestuffs. As another result, gastrointestinal diseases were among the diseases most frequently experienced by the human communities of the İnönü Cave, who were heavily exposed to Cyanobacteria in the Chalcolithic Age and Proteobacteria in the Early Bronze Age.

This study demonstrates the presence of ARGs such as tetA, intl1, and OXA58 in ancient soil samples from İnönü Cave (Fig S1, Fig S2 and Fig S3 in S1 File). This finding suggests that antibiotic resistance existed before using modern antibiotics and has been a longstanding characteristic of microbial communities. The metabarcoding results revealed substantial alterations in the microbial community composition across various historical periods, characterized by significant shifts in the prevalence of phyla such as Proteobacteria and Cyanobacteria. These changes are indicative of the influence of past human activities and environmental conditions. These findings emphasize the intricate nature of the evolution of antibiotic resistance, which is shaped by both natural forces and human actions spanning thousands of years. This research highlights the significance of a comprehensive and interdisciplinary approach that combines archaeological and microbiological perspectives to comprehend the long-term effects of human activities on soil microbiomes and antibiotic resistance. It provides valuable insights into microbial communities' evolutionary history and resistance mechanisms.

This study supported the one-health approach and once again proved the existence of a holistic life activity.

## Supporting information

**S1 File. Agarose raw gel blots electrophoresis image of PCR results for antibiotic resistance genes in soil samples from Inonu Cave.**
(PDF)

**S1 Fig. Agarose raw gel blot electrophoresis image of PCR results for antibiotic resistance genes in soil samples from Inonu Cave.**
(JPG)

**S2 Fig. Agarose raw gel blot electrophoresis image of PCR results for antibiotic resistance genes in soil samples from Inonu Cave.**
(JPG)

**S3 Fig. Agarose raw gel blot electrophoresis image of PCR results for antibiotic resistance genes in soil samples from Inonu Cave.**
(JPG)

## Acknowledgments

The opinions, findings, and conclusions or recommendations expressed in this material are solely the responsibility of the authors and do not necessarily represent the official views of the funding sources.

We present this article to the Great Leader Mustafa Kemal Atatürk and all our martyrs on the 100th anniversary of the Türkiye Republic.

## Author contributions

**Conceptualization:** Sukran Ozturk, F.Gülden Ekmen, Emre Keskin, Benjamin Stanley Arbuckle.

**Data curation:** Sukran Ozturk, F.Gülden Ekmen, Emre Keskin, Benjamin Stanley Arbuckle.

**Formal analysis:** Sukran Ozturk.

**Funding acquisition:** Sukran Ozturk, F.Gülden Ekmen.

**Investigation:** Sukran Ozturk, F.Gülden Ekmen, Hamza Ekmen, Esra Mine Ünal, Ayşegül Er.

**Methodology:** Esra Mine Ünal, Ayşegül Er, Emre Keskin.

**Project administration:** Sukran Ozturk.

**Resources:** Sukran Ozturk, F.Gülden Ekmen.

**Software:** Sukran Ozturk, Hamza Ekmen, Emre Keskin.

**Supervision:** Sukran Ozturk, F.Gülden Ekmen, Hamza Ekmen, Emre Keskin, Benjamin Stanley Arbuckle.

**Validation:** Sukran Ozturk, Hamza Ekmen, Esra Mine Ünal, Ayşegül Er, Emre Keskin.

**Visualization:** Sukran Ozturk, F.Gülden Ekmen, Emre Keskin.

**Writing – original draft:** Sukran Ozturk, F.Gülden Ekmen, Hamza Ekmen, Esra Mine Ünal, Emre Keskin.

**Writing – review & editing:** Benjamin Stanley Arbuckle.

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
