## [Decision Letter · Decision Letter 0]

Dear Dr. Ozturk,

Thank you for submitting your manuscript to PLOS ONE. After careful consideration, we feel that it has merit but does not fully meet PLOS ONE’s publication criteria as it currently stands. Therefore, we invite you to submit a revised version of the manuscript that addresses the points raised during the review process.

Your manuscript has now been seen by a referee, whose comments are appended below. You will see from these comments that while the referee finds your work of potential interest, the reviewer has raised substantial concerns that must be addressed. In light of these comments, we cannot accept the manuscript for publication, but would be interested in considering a revised version that addresses these serious concerns.

We hope you will find the referee's comments useful as you decide how to proceed. Should presentation of further data and analysis allow you to address these criticisms, we would be happy to look at a substantially revised manuscript. However, please bear in mind that we will be reluctant to approach the referees again in the absence of major revisions.

We look forward to receiving your revised manuscript.

Kind regards,

Peter F. Biehl, PhD

Academic Editor

PLOS ONE

Journal Requirements:

2.PLOS ONE now requires that authors provide the original uncropped and unadjusted images underlying all blot or gel results reported in a submission’s figures or Supporting Information files. This policy and the journal’s other requirements for blot/gel reporting and figure preparation are described in detail at https://journals.plos.org/plosone/s/figures#loc-blot-and-gel-reporting-requirements and https://journals.plos.org/plosone/s/figures#loc-preparing-figures-from-image-files. When you submit your revised manuscript, please ensure that your figures adhere fully to these guidelines and provide the original underlying images for all blot or gel data reported in your submission. See the following link for instructions on providing the original image data: https://journals.plos.org/plosone/s/figures#loc-original-images-for-blots-and-gels.   

4. In your manuscript, please provide additional information regarding the specimens used in your study. Ensure that you have reported human remain specimen numbers and complete repository information, including museum name and geographic location. 

For more information on PLOS ONE's requirements for paleontology and archeology research, see https://journals.plos.org/plosone/s/submission-guidelines#loc-paleontology-and-archaeology-research.

5. In your Methods section, please provide additional information regarding the permits you obtained for the work. Please ensure you have included the full name of the authority that approved the field site access and, if no permits were required, a brief statement explaining why.

6. Thank you for stating the following financial disclosure: 

ZBEUN Scientific Research Coordinator (BAP) number 2021-91149634-02.    

7. Thank you for stating the following in the Acknowledgments Section of your manuscript: 

The opinions, findings, and conclusions or recommendations expressed in this material are

solely the responsibility of the authors and do not necessarily represent the official views of the funding sources.

This project was made with the support of ZBEUN BAP number 2021-91149634-02. We

Thanked you ZBEUN Scientific Research Coordinator.

We present this article to the Great Leader Mustafa Kemal Atatürk and all our martyrs on the

occasion of the 100th anniversary of Turkiye Republic.

ZBEUN Scientific Research Coordinator (BAP) number 2021-91149634-02.  

8. We note that your Data Availability Statement is currently as follows: All relevant data are within the manuscript and its Supporting Information files.

9. Please amend your list of authors on the manuscript to ensure that each author is linked to an affiliation. Authors’ affiliations should reflect the institution where the work was done (if authors moved subsequently, you can also list the new affiliation stating “current affiliation:….” as necessary)

10. Please include a separate caption for each figure in your manuscript.

Additional Editor Comments :

Your manuscript has now been seen by a referee, whose comments are appended below. You will see from these comments that while the referee finds your work of potential interest, the reviewer has raised substantial concerns that must be addressed. In light of these comments, we cannot accept the manuscript for publication, but would be interested in considering a revised version that addresses these serious concerns.

We hope you will find the referee's comments useful as you decide how to proceed. Should presentation of further data and analysis allow you to address these criticisms, we would be happy to look at a substantially revised manuscript. However, please bear in mind that we will be reluctant to approach the referees again in the absence of major revisions.

Reviewers' comments:

Reviewer's Responses to Questions

**Comments to the Author**

1. Is the manuscript technically sound, and do the data support the conclusions?

Reviewer #1: Partly

2. Has the statistical analysis been performed appropriately and rigorously?

Reviewer #1: I Don't Know

3. Have the authors made all data underlying the findings in their manuscript fully available?

Reviewer #1: Yes

4. Is the manuscript presented in an intelligible fashion and written in standard English?

Reviewer #1: No

Reviewer #1: Ad 1) The manuscript is partly technically sound as the authors do not comprehensively explain the methodologies and fail to explain why they chose specific tools.

2) There is no description on the statistical methods included in the paper, also no settings and threshold of the employed bioinformatics tools.

3) Yes, it is deposited.

4) The language requires major revision.

**Do you want your identity to be public for this peer review?** For information about this choice, including consent withdrawal, please see our Privacy Policy

Reviewer #1: No

---

## [Author Response · Author response to Decision Letter 1]

18 Mar 2025

Dear Reviewers;

This article brings together archaeology and microbiology and reveals bacterial populations and antibiotic resistance genes identified by examining soil microbiota detected in ancient times, and provides remarkable results in shedding light on the subject of ONE HEALTH. It is an interdisciplinary study covering different topics and raising awareness about antibiotic resistance.

All revisions have been completed in line with your suggestions, and we would like to thank you, Dear Editor, and the referees who contributed to our article. We believe that our article is worthy of publication in PLOS ONE and we kindly request you and the team to give this chance. We believe that this article, which will be a guide for future projects, will also close the gaps in the literature on the subject.

We kindly request that our article be evaluated positively

Respectfully

---

## [Decision Letter · Decision Letter 1]

Dear Dr. Ozturk,

Thank you for submitting your manuscript to PLOS ONE. After careful consideration, we feel that it has merit but does not fully meet PLOS ONE’s publication criteria as it currently stands. Therefore, we invite you to submit a revised version of the manuscript that addresses the points raised during the review process.

Please address the comments in detail before resubmission

We look forward to receiving your revised manuscript.

Kind regards,

Peter F. Biehl, PhD

Academic Editor

PLOS ONE

Journal Requirements:

Additional Editor Comments :

Please address the comments in detail before resubmission

Reviewers' comments:

Reviewer's Responses to Questions

**Comments to the Author**

Reviewer #1: (No Response)

2. Is the manuscript technically sound, and do the data support the conclusions?

Reviewer #1: Yes

3. Has the statistical analysis been performed appropriately and rigorously?

Reviewer #1: Yes

4. Have the authors made all data underlying the findings in their manuscript fully available?

Reviewer #1: Yes

5. Is the manuscript presented in an intelligible fashion and written in standard English?

Reviewer #1: Yes

Reviewer #1: Dear authors,

I appreciate your extensive revision of the manuscript and the incorporation of my comments. I added a few more in my attached reviewer letter but overall am in support of the publication of your manuscript.

**Do you want your identity to be public for this peer review?** For information about this choice, including consent withdrawal, please see our Privacy Policy

Reviewer #1: No

---

## [Author Response · Author response to Decision Letter 2]

22 May 2025

PONE-D-24-53725R1

Decoding Past Microbial Life and Antibiotic Resistance in İnonü Cave’s Archaeological Soil

PLOSONE

Response to Reviewers'.

A marked-up copy of your manuscript that highlights changes made to the original version. You should upload this as a separate file labeled 'Revised Manuscript with Track Changes'.

-Response: - The required document is attached.

An unmarked version of your revised paper without tracked changes. You should upload this as a separate file labeled 'Manuscript'.

-Response : - The required document is attached.

Journal Requirements:

Please review your reference list to ensure that it is complete and accurate. If you have cited retracted articles, please include the reason for doing so in the text of the manuscript or remove these references and replace them with relevant, up-to-date references. Any changes to the reference list should be noted in the rebuttal letter accompanying your revised manuscript. If you must cite a retracted article, indicate the retraction status of the article in the References list and also include a citation and full reference for the retraction notice.

-Response: There have been changes in the references because there were reference additions. All changes are shown in the text, in the references section and on the renamed b31 document.

1.The introduction was well expanded and in its current state adds a lot more relevant context to the study. We would like to highlight again that every statement (especially referring to previous studies) requires a reference. Please revise the text with this in mind. Here are a few examples: The field of ancient DNA (aDNA) and microbial communities in archaeological contexts has received considerable attention in recent years. These investigations offer a distinct viewpoint on human behavior, health, and interactions with the environment in the past. By examining microbial DNA found in archeological sites, scientists can extract significant insights regarding the microbiomes of ancient civilizations, their food patterns, and the occurrence of contagious Illnesses (REF)

-Response: - Appropriate reference added as suggested. (Higuchi, R.; Bowman, B.; Freiberger, M.; Ryder, O.A.; Wilson, A.C. DNA sequences from the quagga, an extinct member of the horse family. Nature 1984, 312, 282–284. [CrossRef] [PubMed] 3. Pääbo, S. Molecular cloning of Ancient Egyptian mummy DNA. Nature 1985, 314, 644–645. [CrossRef] [PubMed]).

2.Furthermore, examining ancient microbial communities allows us to comprehend the historical influence of human actions on environmental microbial ecosystems. Recent interdisciplinary studies have shown that the materials contained in archaeological sediments provide information beyond just defining culture (REF).

-Response: Appropriate reference added as suggested.- Veeramah, K.R. Primate paleogenomics. Paleogenomics Genome-Scale Anal. Anc. DNA 2018, 353–373. [CrossRef]

3.In this context, organic remains dating back thousands of years play an important role. While research on organic remains found in archaeological sites has gained momentum in recent years, the number of studies, especially on microbial diversity and change, still needs to be increased. High-throughput sequencing technologies have opened new frontiers in microbial community analysis and have become widely used in assessing the diversity of bacterial components in soil (REF).

- Response: Appropriate reference added as suggested -Orlando, L.; Allaby, R.; Skoglund, P.; Sarkissian, C.; Stockhammer, P.W.; Ávila-Arcos, M.C.; Fu, Q.; Krause, J.; Willerslev, E.; Stone, A.C.; et al. Ancient DNA analysis. Nat. Rev. Methods Primers 2021, 1, 14. [CrossRef])

4.The recent application of next-generation sequencing methods such as Illumina and Roche provides reliable and accurate resultsin detecting microbial taxa. 16S rRNA studies are preferred as an excellent phylogenetic method to obtain important information about investigating and detecting microbial taxa and antibiotic resistance genes in soil samples from archaeological excavations (REF).

-Response: Appropriate references added as suggested

McVean, G. A Genealogical Interpretation of Principal Components Analysis. PLoS Genet. 2009, 5, e1000686. [CrossRef]

Patterson, N.; Price, A.L.; Reich, D. Population Structure and Eigenanalysis. PLoS Genet. 2006, 2, e190. [CrossRef]

5.Now, more than three decades later, scientists and researchers are actively engaged in ongoing research to solve various puzzles related to the origin of humanity, migration patterns, the emergence of infectious diseases, and their spread among ancient populations. aDNA analysis has emerged as a state-of-the-art genetic tool gaining momentum worldwide, changing our comprehensive understanding of the past. For example, in a recent groundbreaking study, aDNA analysis of seven individuals revealed the origin of the plague strain that caused the Black Death in present-day (2) (3). Are you sure (2) is the appropriate reference here?

-Response: Necessary investigations were made and reference number 2, Dalal V, Pasupuleti N, Chaubey G, Rai N, Shinde V. Advancements and Challenges in Ancient DNA Research: Bridging the Global North–South Divide. Genes. 2023;14(2):479. was confirmed to be relevant to this topic. However, since references were added to previous sections based on your suggestions, its number was changed to ... .

-Other added references were marked and indicated in the text and references section.

6.More broadly, aDNA data have revolutionized our knowledge and curiosity and have led to the publication of an enormous amount of literature. The latest technological innovations (If you mention innovations, make sure to name them! have proven to be an extraordinary tool for scientists and researchers in the race to obtain the 'gold' (aDNA) I would paraphrase this or leave it out as it is neither common nor scientific to refer to aDNA as “gold”.. In the present research, we aim to broaden the scope of the ongoing conversation in the field of aDNA by reporting relevant literature published worldwide and detailing advances and challenges.(Put aims at the end of the introduction rather than in the middle. Further, it seems obsolete to mention you’re including a state of the art literature review as this is just common scientific practice in a paper. I would remove this sentence.)

-Response: It has been reviewed in line with your suggestions and these statements have been removed upon your suggestion.

7. While significant microbial abundance and diversity were found in soil samples from four archaeological levels, our study does not reveal differences in microbial community structure and diversity. These differences may be related to various consumer, disease, and nutritional factors. These microarchaeological findings and microbial characteristics may be attributed to different microbial responses to specific consumer habits. Hence, different consumption habits, patterns, and disease profiles lead to different compositions of the generated wastes and thus leave different microbial traces.

This statement includes results and conclusions and does not really fit in the introduction.

- Response: This section has been moved to the results section for your recommendations and manuscript layout.

8.Site (İnönü Cave) and Samples

İnönü Cave is a cave of volcanic origin located in the Cambu geological formation. This volcanic cave, located on the Western Black Sea coast of Turkiye, developed in several stages (Fig 1) 5

Fig 3: Number of Identified Specimens (%) (17)

Why is there a change of reference style (superscript)?

-Response: (Fig 1)5

Fig 3: Number of Identified Specimens (%) (17)

-Reference numbers 5 and 17 have been converted from superscript to normal notation.

9. Bioinformatics analysis identified the presence of a wide range of taxa, including Cyanobacteria, the Acidobacteriota phylum, the Actinobacteriota phylum, Bacteroidota, Chloroflexi, Firmicutes, Myxococcota, Proteobacteria, Cyanobacteria, and Nitrospirota species

(Tabel 2). Typo: Table

-Response: (Tabel 2). Typo: Table expression has been updated to Table.

10. Material and Method (Rather use the plural Materials and Methods)

-Response : It has been corrected as Materials and Methods

11. Assoc analyzed the defined fillings analogically. Prof. Dr. Hamza Ekmen and Assoc. Prof. Dr. F. Gülden Ekmen and confirmed with 14 C analyses, were taken with sterile materials (Eppendorfs, falcon tubes, steel spatulas, steel collecting spoons) by pharmaceutical microbiologist Assoc. Prof. Dr. Sukran Ozturkfrom the Faculty of Pharmacy. (Paraphrase these sentences, the structure is not correct) The materials were sterilized in an autoclave (brand Nüve) at 121° C for 15 minutes in the Pharmaceutical Microbiology Laboratory of the Beun Faculty of Pharmacy and transported to the excavation site (Figures 3-4).

-Response: This paragraph has been revised as follows.

The fillings of the four archaeological levels were verified with C14 carbon analyses and evaluated by Assoc. Prof. Dr. Hamza Ekmen and Assoc. Prof. Dr. F. Gülden Ekmen.

Soil samples were taken from these layers under the control of the excavation head Assoc. Prof. Dr. Hamza Ekmen and the responsible researcher Assoc. Prof. Dr. Gülden Ekmen, using materials prepared by Beun Faculty of Pharmacy, Department of Pharmaceutical Microbiology, Head of the Department of Pharmaceutical Microbiology, Assoc. Prof. Dr. Şükran Öztürk, sterilized in an autoclave (Nüve) at 121° C for 15 minutes in the Pharmaceutical Microbiology Laboratory of Beun Faculty of Pharmacy, and considering antisepsis conditions (Eppendorfs, falcon tubes, steel spatulas, steel collecting spoons) and transferred to the laboratory environment under sterile conditions (Figure 3-4).

12. Detailed information is shown in Table 1. (I don’t hink the NISP Figure 3 is relevant here)

PCR and Bioinformatics Small 16S gene region primers (16SV3-F/16SV3-R) were used for the qPCR test to see if PCR inhibition happened 15 . Again superscript.

-Response: The superscript has been corrected to normal spelling. (15)

13. Data availability

Our data falls under the list of data types that must be deposited with you. We provide the relevant accession numbers, and the name of the database in which your data is stored is "The datasets generated and/or analyzed during the current study are available in the NCBI BioProject repository, BioProcecj ID: PRJNA1134133."

Omit the first part, this is not necessary to mention and just state: “The datasets generated and/or analyzed during the current study are available in the NCBI BioProject repository, BioProcecj ID: PRJNA1134133.”

-Response: The specified corrections were made in accordance with the suggestions. The name of the database in which your data is stored is “The datasets generated and/or analysed during the current study are available in the NCBI BioProject repository, BioProcecj ID: PRJNA1134133.”

Results

14. We have found information supporting our study on these taxa (Fig 6a,b,c,d). The figures are not supporting this statement but your own results. Leave this sentence out. You are discussing your results in a later part of the results anyways.

-Response: We have found information supporting our study on these taxa (Fig 6a,b,c,d). “This sentence has been removed based on your suggestions.

15. The MN NUCLEOSPIN Soil DNA Extraction Kit was chosen for its proven efficiency in isolating ancient DNA (aDNA) from complex soil matrices, which often contain high levels of PCR inhibitors such as humic acids. This kit's robust inhibitor removal technology ensures higher purity and yield of DNA, which is critical for downstream applications in ancient microbiome studies. Furthermore, its dual-buffer system (SL1 and SL2) optimized lysis efficiency across varied soil types, enhancing DNA recovery from both Gram-positive and Gram-negative bacteria (Fig 6a,b,c,d).

This is a repetition form the Methods section. Shorten or leave out.

Following this paragraph there is an icon of a file named “aDNA_silva_krona.html” in the text. This seems to be a mistake.

-Response: This sentence has been removed based on your suggestions. “aDNA_silva_krona.html”

-This link was shared so that you could access the metagenomic data. However, it was removed based on your suggestions since it was not possible to access it.

16. The Actinobacteriota phylum, or Actinobacteria, is a diverse group of bacteria found in various environments, including soil, water, and plants. They are characterized by their high G+C content in their genomic DNA and filamentous growth pattern. Actinobacteria are known for their ability to produce a wide range of bioactive compounds, such as antibiotics, antifungals, and antitumor agents. This has led to extensive research into their potential applications in medicine and biotechnology. Additionally, Actinobacteria play essential roles in soilecosystems, involving nutrient cycling and organic matter degradation (51).

Another identified bacterial taxon is Bacteroidota. They are commonly found in aquatic environments, soil, and the guts of animals, including humans. Bacteroidota bacteria are essential in breaking down complex organic matter, such as polysaccharides, and play a vital role in the carbon cycle (25).

These detailed descriptions of the phyla seems a bit repetitive as well. Shorten and focus on your

results.

-Response: This section has been shortened according to recommendations.

17. Lactobacillus species produce fermented foods such as yogurt and cheese (27, 28). Bacillus species are known for making antibiotics and enzymes, while Clostridium species are essential in producing biogas and other industrial products. Lactobacillus species produce fermented foods such as yogurt and cheese, while Staphylococcus aureus can cause human infections (27, 28).

Repetitive.

-Response: Lactobacillus species produce fermented foods such as yogurt and cheese, while Staphylococcus aureus can cause human infections

-This section is edited according to recommendations.

Note: The quality of the figures, even after downloading them, is so bad that they could not be evaluated by us.

-Response: The quality of the figures has been revised.

AConclusio:

This revision added substantial value to the manuscript and after addressing our aforementioned comments, we recommend publication.

- Thank you for your consideration. Respectively

Assos.Prof. Sukran OZTURK PhD

---

## [Editor Report · Decision Letter 2]

Decoding Past Microbial Life and Antibiotic Resistance in İnonü Cave’s Archaeological Soil

PONE-D-24-53725R2

Dear Dr. Ozturk,

We’re pleased to inform you that your manuscript has been judged scientifically suitable for publication and will be formally accepted for publication once it meets all outstanding technical requirements.

Kind regards,

Peter F. Biehl, PhD

Academic Editor

PLOS ONE
---

## [Editor Report · Acceptance letter]

PONE-D-24-53725R2

PLOS ONE

Dear Dr. Ozturk,

I'm pleased to inform you that your manuscript has been deemed suitable for publication in PLOS ONE. Congratulations! Your manuscript is now being handed over to our production team.

Kind regards,

on behalf of

Dr. Peter F. Biehl

Academic Editor

PLOS ONE